# Molecular Epidemiology of Multidrug-Resistant Uropathogenic *Escherichia coli* O25b Strains Associated with Complicated Urinary Tract Infection in Children

**DOI:** 10.3390/microorganisms9112299

**Published:** 2021-11-05

**Authors:** Laura M. Contreras-Alvarado, Sergio Zavala-Vega, Ariadnna Cruz-Córdova, Juan Pablo Reyes-Grajeda, Gerardo Escalona-Venegas, Víctor Flores, Virginia Alcázar-López, José Arellano-Galindo, Rigoberto Hernández-Castro, Graciela Castro-Escarpulli, Juan Xicohtencatl-Cortes, Sara A. Ochoa

**Affiliations:** 1Laboratorio de Investigación en Bacteriología Intestinal, Unidad de Investigación en Enfermedades Infecciosas, Hospital Infantil de México Federico Gómez, Ciudad de México 06720, Mexico; 20.93lau@gmail.com (L.M.C.-A.); ariadnnacruz@gmail.com (A.C.-C.); gesven2000@hotmail.com (G.E.-V.); 2Laboratorio de Investigación Clínica y Ambiental, Escuela Nacional de Ciencias Biológicas, Instituto Politécnico Nacional, Ciudad de México 11340, Mexico; chelacastro@hotmail.com; 3Laboratorio de Neuropatología, Instituto Nacional de Neurología y Neurocirugía Manuel Velasco Suárez, Ciudad de México 14269, Mexico; sergio.zavala.vega@gmail.com; 4Subdirección de Desarrollo de Aplicaciones Clínicas, Instituto Nacional de Medicina Genómica, Ciudad de México 14610, Mexico; jreyes@inmegen.gob.mx; 5Department of Biochemistry, University of Cambridge, 80 Tennis Court Road, Cambridge CB2 1GA, UK; vf28@cam.ac.uk; 6Departamento de Laboratorio Clínico, Laboratorio Central, Hospital Infantil de México Federico Gómez, Ciudad de México 06720, Mexico; qfbalcazarclinicos@gmail.com; 7Laboratorio de Virología Clínica y Experimental, Unidad de Investigación en Enfermedades Infecciosas, Hospital Infantil de México Federico Gómez, Ciudad de México 06720, Mexico; jose.arellano@salud.gob.mx; 8Departamento de Ecología de Agentes Patógenos, Hospital General Dr. Manuel Gea González, Ciudad de México 4800, Mexico; rigo37@gmail.com

**Keywords:** UPEC O25, MDR, MLST, genetic diversity

## Abstract

Background: Uropathogenic *Escherichia coli* (UPEC) has increased the incidence of urinary tract infection (UTI). It is the cause of more than 80% of community-acquired cystitis cases and more than 70% of uncomplicated acute pyelonephritis cases. Aim: The present study describes the molecular epidemiology of UPEC O25b clinical strains based on their resistance profiles, virulence genes, and genetic diversity. Methods: Resistance profiles were identified using the Kirby–Bauer method, including the phenotypic production of extended-spectrum β-lactamases (ESBLs) and metallo-β-lactamases (MBLs). The UPEC serogroups, phylogenetic groups, virulence genes, and integrons were determined via multiplex PCR. Genetic diversity was established using pulsed-field gel electrophoresis (PFGE), and sequence type (ST) was determined via multilocus sequence typing (MLST). Results: UPEC strains (*n* = 126) from hospitalized children with complicated UTIs (cUTIs) were identified as O25b, of which 41.27% were multidrug resistant (MDR) and 15.87% were extensively drug resistant (XDR). The O25b strains harbored the *fimH* (95.23%), *csgA* (91.26%), *papG*II (80.95%), *chuA* (95.23%), *iutD* (88.09%), *satA* (84.92%), and *intl1* (47.61%) genes. Moreover, 64.28% were producers of ESBLs and had high genetic diversity. ST131 (63.63%) was associated primarily with phylogenetic group B2, and ST69 (100%) was associated primarily with phylogenetic group D. Conclusion: UPEC O25b/ST131 harbors a wide genetic diversity of virulence and resistance genes, which contribute to cUTIs in pediatrics.

## 1. Introduction

Urinary tract infections (UTIs) are a public health problem in Mexico and the second leading cause of morbidity worldwide. UTIs are the main infections associated with prolonged hospital stay. The use of medical devices, such as urinary catheters, has increased the incidence of UTIs of hospital origin, which are primarily associated with uropathogenic *Escherichia coli*, commonly called UPEC [1].

UPEC is the causative agent of more than 80% of community-acquired cystitis, more than 70% of uncomplicated acute pyelonephritis, and 3.6–12.6% of complicated UTIs (cUTIs) with urosepsis [2]. Serological typing using serogroup O (lipopolysaccharide) has made it possible to identify clinical strains of UPEC with well-identified virulence factors [3]. Clinical strains of UPEC are primarily related to serogroups O1, O2, O4, O6, O7, O8, O15, O16, O18, O21, O22, O25, O75, and O83 [3]. Clinical strains of UPEC belonging to serogroup O25 (UPEC O25) have been reported in various populations, including adult patients hospitalized with symptomatic UTIs [4], adult patients with community-associated UTIs [5,6], female patients prone to acute cystitis [7], and children with UTIs [8].

Clinical strains of UPEC O25 are related to a high presence of virulence factors, such as fimbrial adhesins (FimH, PapG, and SfA); toxins such as cytotoxic necrotizing factor 1 (CNF1), α-hemolysin (HlyA), and enterotoxin (Set1); iron scavengers (IronN); and siderophores (Iuc and FyuA) [4,5]. UPEC O25 clinical strains have a multidrug-resistant (MDR) profile for various antibiotics and produce extended-spectrum β-lactamases (ESBLs) [9].

Phylogenetic tools for the identification of clonal complexes (CCs) and sequence types (STs) differentiate UPEC strains based on their pathogenic and commensal origins. CC73, CC69, and CC141 were associated with UPEC strains recovered from patients with UTIs, and the CC10 and CC95 complexes were found primarily among commensal strains of the normal intestinal microbiota of healthy individuals [10]. However, the identification of CCs in pathogenic and commensal strains from healthy individuals is indicative of cross contamination during the establishment of UTIs [10]. Several STs (ST69, ST10, ST131, and ST648) were identified in extraintestinal strains of *E. coli*. The typification in specific populations of STs is primarily related to geographic region [11,12,13]. UPEC O25b clinical strains from patients with UTIs belonging to ST131 were related to hospital outbreaks, had a multidrug-resistant (MDR) lineage, and were ESBL producers of the cefotaximase-M (CTX-M) type, which hydrolyze third-generation cephalosporins (cefotaxime, ceftriaxone, and ceftazidime) [14].

Clinical strains of MDR and extremely drug-resistant (XDR) UPEC isolated from Mexican children with cUTIs in the Children’s Hospital of Mexico Federico Gomez (HIMFG) were previously characterized [15,16]. Phylogenetic analyses using pulsed-field gel electrophoresis (PFGE) and enterobacterial repetitive intergenic consensus–polymerase chain reaction (ERIC-PCR) showed the presence of clades that were closely related to phylogenetic groups B2 and D [15,16]. Preliminary data described UPEC strains collected from pediatric patients of the HIMFG with an MDR profile and a serogroup of O25b [16]. STs and CCs were related to virulence in clinical strains of UPEC O25b from pediatric patients, but additional studies are required to elucidate their pathogenesis. In the present study, we evaluated the resistance profile against 10 antibiotic categories, β-lactamase phenotype expression [ESBL and Metallo-β-Lactamase (MBL)], integron classes (1 and 2), and frequency of 17 virulence genes to analyze genetic diversity using PFGE (macrorestriction patterns) and multilocus sequence typing (MLST; STs and CCs) in a set of clinical strains of serogroup O25b UPEC recovered from children with cUTIs from the HIMFG.

## 2. Materials and Methods

### 2.1. Clinical Strains of UPEC

A total of 129 clinical UPEC strains of serotype O25b were included in this study. The UPEC strains were recovered from pediatric patients with cUTIs who were hospitalized in different service areas of HIMFG from 2009 to 2018. The clinical UPEC strains were phenotypically identified using an automated Vitex^®^ 2 system (BioMérieux, Marcy l’Étoile, France) and kept at −70 °C in brain heart infusion broth (BHI; BD-Difco, Franklin Lakes, NJ, USA) supplemented with 20% glycerol (Sigma-Aldrich, St. Louis, MO, USA) and 0.5% fetal bovine serum (ATCC, Manassas, VA, USA).

### 2.2. Typing of UPEC O25b Clinical Strains

Genomic DNA was extracted using a Quick-DNA^TM^ Universal Kit (Zymo Research, Irvine, CA, USA) and quantified in a NanoDrop^TM^ 2000 (Thermo Fisher Scientific, Waltham, MA, USA). Genomic DNA was electrophoretically separated to verify its integrity and used for the identification of UPEC strains of serotype O25b by allele-specific double PCR [17]. Briefly, PCR amplified the 347 bp *pabB* gene, which is specific for serogroup O25b/ST131 due to a single-nucleotide polymorphism (SNP) at the 3′ position, and the 427 bp *trpA* gene, which is a housekeeping gene (tryptophan synthase alpha subunit). The specific primers and amplification conditions are shown in Appendix A. PCR was performed with MasterMix (Promega, Madison, WI, USA) and 1 µL of DNA (200 ng/µL) in a total reaction volume of 12.5 µL under the following conditions: denaturation at 95 °C for 5 min; 35 cycles of denaturation at 95 °C for 1 min, annealing at 65 °C for 1 min, and extension at 72 °C for 1.5 min; and final extension at 72 °C for 5 min. The DNA products were verified by electrophoresis in 1% agarose gels after staining with 0.5 mg/mL ethidium bromide (Sigma Aldrich, St. Louis, MO, USA) and visualization under UV light using a ChemiDoc MP imaging system (Bio-Rad, Hercules, CA, USA). UPEC 42U-0612 O25b was used as a positive control [16].

### 2.3. Antimicrobial Susceptibility Profile of UPEC O25b Strains

The susceptibility to antibiotics of the 129 clinical strains of UPEC O25b was determined using the Kirby–Bauer method via diffusion with Sensi-Discs (BD, Franklin Lakes, NJ, USA) on Mueller-Hinton agar (MHA, BD-Difco, Franklin Lakes, NJ, USA). Seventeen antibiotics from 10 categories were included in this study: penicillins: ampicillin (AM-10 µg); β-lactam combination agents: amoxicillin–clavulanate (AMC-20/10 µg) and piperacillin–tazobactam (TZP-100/10 µg); first- and second-generation cephems: cephalothin (CF-30 µg) and cefaclor (CEC-30 µg); third-generation cephems: cefotaxime (CTX-30 µg) and ceftazidime (CAZ-30 µg); monobactams: aztreonam (ATM-30 µg); fluoroquinolones: levofloxacin (LVX-5 µg), norfloxacin (NOR-10 µg), and ofloxacin (OFX-5 µg); carbapenems: meropenem (MEM-10 µg) and imipenem (IPM-10 µg); aminoglycosides: gentamicin (GM-10 µg); folate pathway antagonists: trimethoprim–sulfamethoxazole (SXT-1.25/23.75 µg); tetracyclines: tetracycline (TE-30 µg); and nitrofurans: nitrofurantoin (F/M-300 µg). The methodology was performed following the indications of the Clinical and Laboratory Standards Institute (CLSI) [18] and as described by Ochoa et al. [15]. MDR strains were defined as strains with acquired nonsusceptibility to at least one antibiotic in three or more antibiotic categories. XDR strains were defined as strains that were not susceptible to at least one agent in all but two or fewer antibiotic categories [19].

### 2.4. Phenotypic Determination of ESBL and MBL Production in Clinical Strains of UPEC O25b

The phenotypic determination of ESBL and MBL production was performed using sensitivity tests to several antibiotic discs: for ESBLs, CAZ (30 µg), ATM (30 µg), and CRO (30 µg); and for MBLs, MEM (10 µg) and IPM (10 µg). Phenotypic determinations were performed as suggested by the CLSI [18]. Double-disc synergy and individual disc assays were performed as previously described by our working group [15]. The ESBL phenotype of clinical UPEC strains was also confirmed using Hodge’s test. *Klebsiella pneumoniae* ESBLs+ (ATCC 700603), *E. coli* ESBLs negative (ATCC 25922), *Pseudomonas aeruginosa* MBLs− (ATCC 27853), and *P. aeruginosa* MBLs positive (540UC1) were used as controls [15,20].

### 2.5. Typing of Phylogenetic Groups, Virulence Genes, and Integrons Associated with UPEC O25b Strains

Phylogenetic groups and subgroup typing for the clinical strain UPEC O25b were performed as proposed by Clermont et al. [21]. The UPEC 42U-0612 strain belonging to phylogenetic group B2_2_ was used as a positive control [16].

Seventeen genes associated with UPEC pathogenesis were investigated: three variant adhesins of the PapG pilus (*papG*I, *papG*II, and *papG*III), the FimH adhesin of the type 1 fimbria (*fimH*), the CsgA adhesin of fimbria curli (*csgA*), the EcpA adhesin of the ECP pilus (*ecpA*), cellulose (a surface glycan, *bcsA*), the IutD Fe-binding siderophore aerobactin (*iutD*), yersiniabactin (*fyuA*), the ChuA heme group scavenger (*chuA*), the HlyA pore-forming toxin (*hlyA),* the TosA RTX protein (*tosA*), the SatA autotransporter (*satA*), cytotoxic necrotizing factor 1 (*cnf1*), and flagellar proteins [flagellin *fliC*) and flagellar motor (*motA* and *motB*)]. Endpoint and multiplex PCRs were performed in a Veriti 96-well thermal cycler (Applied Biosystems, Foster City, CA, USA). UPEC CFT-073 and previously reported clinical strains were included in this study and used as reference controls [15,16]. The class 1 (*intl1*) and 2 (*intl2*) integrase genes were amplified using multiplex PCR as described by Ochoa et al. [15]. UPEC clinical strains 502U1-0412 (*intl1*) and 674U-0612 (*intl2*) were used as controls [15]. The primers and reaction conditions are shown in Appendix A.

### 2.6. Pulsed-Field Gel Electrophoresis (PFGE) of Clinical UPEC O25b Strains

PFGE tests were performed according to Ochoa et al. [15]. First, the UPEC O25b strains were embedded in low-melting-point (LMP) agarose blocks from Promega Corporation (Madison, WI, USA). Next, digestion of the samples was performed using 20 U of the restriction enzyme Xba1 (New England Biolabs, Ipswich, MA, USA) at 37 °C for 20 h. The PFGE shift was performed for 24 h at 200 V (7 v/cm) to an angle of 120° at 14 °C, with an initial pulse of 2.16 s and a final pulse of 13.58 s, in a CHEF Mapper system (Bio-Rad Life Science Research, Hercules, CA, USA). The macrorestriction products in the PFGE gels were stained with GelRed^®^ Nucleic Acid Stain (Biotium, Fremont, CA, USA), visualized with UV light, and digitized using the CCD Camera Documenting System BK04S-3 (Biobase, Mexico City, Mexico). PFGE pulsotypes were analyzed using NTSYSpc v2.02j and PAST v4.03 software, and the clonality degree was estimated according to the criteria described by Tenover et al. [22]. A lambda ladder PFGE marker (New England Biolabs, Hertfordshire, England, UK) was used as a molecular weight marker in the PFGE assay.

### 2.7. Typing of UPEC O25b Clinical Strains Using MLST

Identification of the STs of the UPEC O25b clinical strains was performed via the amplification of seven housekeeping genes, as suggested by the *Escherichia coli* MLST website (https://pubmlst.org/escherichia/; accessed on 13 October 2021) of the University of Oxford and according to the protocols described by Wirth et al. [23] and Jolley et al. [24]. Briefly, internal fragments of *adk* (adenylate kinase), *fumC* (fumarate hydratase), *gyrB* (DNA gyrase), *icd* (isocitrate/isopropylmalate dehydrogenase), *mdh* (malate dehydrogenase), *purA* (adenylosuccinate dehydrogenase), and *recA* (ATP/GTP binding motif) were designed, amplified (Phusion^TM^ high-fidelity DNA polymerase, (Thermo Fisher Scientific, Waltham, MA, USA), purified, and sequenced using the Illumina NextSeq600 platform (NGS, Illumina, Inc., San Diego, CA, USA) on 1 μg of genomic DNA. Multiple sequence alignment was performed to identify regions with 100% identity. The readings, filtration, assembly, and mapping were analyzed as described by Mancilla-Rojano et al. [25]. The ST for each allelic profile was assigned based on the MLST platform at https://pubmlst.org/escherichia/; accessed on 13 October 2021. The designed primers, their products, and amplification conditions are shown in Appendix A.

### 2.8. Statistical Analysis

ANOVA with Fisher’s protected least significant difference (PLSD) and StatView 4.5 software for Windows (www.statview.com; accessed on 26 September 2019) [26] were used to determine the statistical significance of the data. A *p*-value of <0.05 was considered to indicate statistical significance. A heat map associated with the phylogenetic groups and virulence genes was generated using GraphPad Prism 8.0.1 software (https://www.graphpad.com/scientific-sofware/prism/; accessed on 15 November 2020). PFGE pulsotypes were analyzed with NTSYSpc v2.02j software (Applied Biostatistics, Inc., Setauket, NY, USA) using Jaccard’s coefficient of similarity and clustering analysis by the unweighted pair group method using arithmetic averages (UPGMA) and PAST v4.03 software (https://past.en.lo4d.com/windows; accessed on May 30, 2021). The dendrogram was edited using the iTOL interactive tree of life (https://itol.embl.de/; accessed on 3 June 2021). The CCs were analyzed with PHYLOViZ software (https://online.phyloviz.net/index; accessed on 13 September 2021) using the eBURST algorithm [27].

## 3. Results

### 3.1. UPEC O25b Strains Associated with cUTIs

A total of 129 clinical strains of UPEC O25b were recovered from patients with cUTIs who were hospitalized in various service areas of the HIMFG from 2009 to 2018. The clinical strains of UPEC O25 belonged to ST131-amplified DNA products of length 347 bp and 427 bp, which correspond to the molecular sizes of the *pabB* and *trpA* genes, respectively.

### 3.2. Multidrug Resistance Profile of UPEC O25b Clinical Strains

The antibiotic susceptibility tests revealed that 81.74% (103/126) of the UPEC O25b strains were classified as (MDR), and 18.25% (23/126) were classified as (XDR). The remaining three UPEC O25b strains that were not classified as MDR or XDR were excluded from this study. The 126 UPEC O25b MDR and XDR strains showed resistance to penicillins [AM (99.20%)]; β-lactams combined with inhibitors [TZP (92.85%) and AMC (77.7%)]; first-, second-, and third-generation cephalosporins [CF (85.71%), CEC (83.33%), CTX (77.77%), and CAZ (77.77%)]; and fluoroquinolones [LVX (77.77%), NOR (77.77%), and OFX (77.77%)] (Table 1 and Figure 1). The UPEC O25b strains showed the least resistance to carbapenems [MEN (7.94%) and IPM (7.94%)] and nitrofurans [F/M (7.94%)] (Table 1 and Figure 1). The MDR and XDR profiles of the 126 UPEC O25b strains that showed resistance exhibited the following distribution: 1.58% (2/126) were resistant to three categories of antibiotics (MDR-3), 11.90% (15/126) were MDR-4, 11.11% (14/126) were MDR-5, 15.87% (20/126) were MDR-6, and 41.27% (52/126) were MDR-7. A total of 15.87% (20/126) of the UPEC O25b strains were resistant to eight categories of antibiotics (XDR-8), and 2.38% (3/126) were XDR-9. The MDR-7 phenotype was the most prevalent among the UPEC O25b strains.

### 3.3. Resistance Associated with the Phenotypic Expression of ESBLs and Integrons

ESBL phenotypic expression was determined for the 126 UPEC O25b strains as suggested by CLSI [18]. Briefly, 64.28% (81/126) of the UPEC O25b strains were ESBL producers according to the synergism test (double disc, individual disc with or without clavulanic acid) and Hodge’s test (Table 1 and Figure 1). In contrast, no UPEC O25b strains producing MBLs were identified with the three tests used.

A total of 47.61% (60/126) of the MDR and XDR UPEC O25b strains were positive for class 1 integrons (Table 1), and only 38.33% (23/60) amplified a DNA amplicon of 1800 bp of the variable region. Only 3.17% (4/126) of the MDR and XDR UPEC O25 strains were positive for class 2 integrons, but only two strains amplified a DNA amplicon of 2000 bp of the variable region (Table 1).

### 3.4. Virulence Gene and Phylogenetic Group Associations of UPEC O25b Clinical Strains

MDR and XDR UPEC O25b strains were primarily associated with two subgroups belonging to phylogenetic group B2: 60.31% (76/126) were distributed to subgroup B2_2_ and 9.52% (12/126) were distributed to subgroup B2_1_ (Figure 1). For phylogenetic group D, 14.28% (18/126) were distributed in subgroup D_1_, and 11.11% (14/126) were distributed in subgroup D_2_ (Figure 1). A total of 4.76% (6/126) of the MDR and XDR UPEC O25b strains were grouped into phylogenetic subgroup A1 (Figure 1).

On the other hand, 95.23% (120/126) of the UPEC O25b strains carried the *fimH* gene, 91.26% (115/126) carried the *csgA* gene, 100% (126/126) carried the *ecpA* gene, and 10.31% (13/126) carried the *bcsA* gene. The *papG*II gene of PapG fimbrial adhesin was the most prevalent variant, and it was found in 80.95% (102/126) of the strains, followed by the *papG*III gene, which was found in 2.38% (3/126). Moreover, 99.20% (125/126) carried the *motA* gene, and 99.20% (125/126) carried the *motB* gene. Both of these genes encode flagellar motor proteins. In contrast, 31.74% (40/126) carried the *fliC* gene, which encodes a structural protein of the flagellar filament. Other genes were also identified using multiplex PCR: 95.23% (120/126) carried the *chuA* gene (coding for heme-binding proteins); 88.09% (111/126) carried the *iutD* gene (coding for aerobactin); and 34.92% (44/126) carried the *fyuA* gene (coding for yersiniabactin) (Figure 1). Genes encoding toxins were distributed as follows: 84.92% (107/126) carried the *satA* gene (coding for a secreted autotransporter toxin); 40.47% (51/126) carried the *hlyA* gene (coding for α-hemolysin); 13.49% (17/126) carried the *tosA* gene (coding for the TosA protein); and 33.33% (42/126) carried the *cnf1* gene (coding for the cytotoxic necrotizing factor) (Figure 1).

### 3.5. PFGE Genetic Diversity of UPEC O25b Strains

Dendrogram analysis generated with NTSYSpc and PAST v4.0 and edited with the iTOL Interactive Tree of Life showed high genetic diversity, with 37% similarity among all UPEC O25b strains (Figure 2). In summary, 124 different pulsotypes were identified among the 126 strains with macrorestriction profiles of 11 to 24 bands. According to the analysis of the phylogenetic groups and virulence factors of the MDR and XDR UPEC O25b strains, 31.74% (40/126) of these strains showed a similarity of ≥80%, which indicates a highly related profile (Figure 2).

### 3.6. STs and CCs Maintain an Association with UPEC O25b Clinical Strains

From analyses of STs and CCs, we selected 55 clinical strains of UPEC O5b based on their resistance profiles, virulence, and clonality. Analysis of the STs revealed that 25.45% (14/55) of the UPEC O25b strains belonging to phylogenetic group B2, with an MDR or XDR profile, showed ST131. Furthermore, 21.81% (12/55) of UPEC O25b strains belonging to phylogenetic group D were associated with ST69, which was the most prevalent. However, UPEC O25b/ST131 strains were distributed among the five phylogenetic subgroups B2_1_, B2_2_, D_1_, D_2_, and A_1_. The MLST analysis of the UPEC O25b strains identified the following STs: ST73, ST405, ST93, ST394, ST10, ST120, ST62, ST117, ST443, ST998, ST421, and ST1177 (Appendix A).

UPEC O25b strains grouped in phylogenetic group B2 were distributed primarily in CC131, and strains grouped in phylogenetic group D were distributed in CC69. The UPEC O25b MDR and XDR strains were distributed primarily in CC73, CC405, CC168, CC10, CC205, CC95, CC394, and CC38 (Figure 3 and Appendix A). The phylogenetic analysis of the CCs by eBURST showed a great dispersion of the CCs associated with UPEC O25b strains (Figure 3).

## 4. Discussion

UPEC O25b/ST131 strains are associated with community-acquired and nosocomial UTIs in Mexico. These strains have a high resistance profile to several antibiotics that are commonly used in UTI treatment and a wide repertoire of virulence genes that contribute to UTI pathogenesis [28,29]. UPEC O25b/ST131 strains were successfully identified using the amplified *pabB* gene [29,30,31,32]. Duplex PCR identification of the *pabB* and *trpA* genes facilitated the selection of MDR and XDR UPEC clinical strains of serogroup O25b in this work.

The present study identified 20.25% (126/622) of the UPEC strains from pediatric patients with cUTIs as belonging to serogroup O25. Community-acquired (22.8%) and hospital-acquired (55.6%) UTIs were associated with UPEC O25b/ST131 strains in Istanbul, Turkey [33]. Ten percent of MDR strains were found to belong to the *E. coli* O25b/ST131 clonal group in Kuwait and represent a major public health risk [30]. A total of 44.23% of UPEC O25b/ST131 strains were associated with extraintestinal infections in Egypt [32]. UPEC O25b/ST131 strain producers of ESBLs from UTI pediatric patients have continuously and significantly increased in prevalence in Taiwan, from 33% (2009) to 43% (2010), 74% (2011), and 68% (2012) [34]. Twenty-five percent of UPEC O25b/ST131 strains were associated with the ESBL phenotype in Mexico [28]. However, few reports have characterized large populations of UPEC O25 strains in pediatric patients in Mexico. The carbapenemase-producing O25b/ST131 clone was previously related to health-care-associated infections in the Mexican pediatric population [35].

Among the UPEC strains classified as O25b from children, 81.74% were MDR and 18.25% were XDR. The multidrug resistance exhibited by *E. coli* O25b/ST131 strains in extraintestinal infections was primarily associated with ESBL production of the CTX-M type and fluoroquinolone resistance [28,36]. Our data showed high rates of resistance to different antibiotic categories, including penicillins (99.20%); β-lactams combined with β-lactamase inhibitors (92.85%); first-, second-, and third-generation cephalosporins (85.71%); fluoroquinolones (77.77%); and ESBL phenotypic expression (64.28%). Similarly high rates of resistance and ESBL production were described in other studies [28,30,33,36,37]. In contrast, UPEC O25b strains were most susceptible to carbapenem antibiotics, MEM (7.94%), IPM (7.94%), and F/M (7.94%). Carbapenem and nitrofuran antibiotics are commonly used in the treatment of cUTIs, afebrile UTIs, and UTIs caused by ESBL-producing strains [33,38,39]. However, the guidelines for UTI treatment must be constantly revised for better antibiotic selection during routine use in children, and adequate epidemiological surveillance must be performed [40,41].

Three UPEC O25b strains (532U-O25b, 289U-O25b, and 608U-O25b) were resistant to one or two antibiotics, and these strains were associated with phylogenetic groups B2 and D1. Antibiotic susceptibility was primarily associated with resistance to AM and CF (first generation), CEC (second generation), and CTX (third generation) cephalosporins. However, the phenotypes of ESBL production and integron presence (classes 1 and 2) were not identified. These two mechanisms of resistance spread significantly contributed to the presence of MDR UPEC strains in Mexico [15,16]. UPEC strains were characterized via the identification of several adhesion genes (*fimH*, *csgA*, *papG*II or *papG*III, and *ecpA*), siderophore and iron scavenger genes (*iutD*, *fyuA*, and *chuA*), flagellar genes (*motA*, *motB*, and *fliC*), and toxin-coding genes (*satA*). Our data suggest that these sensitive UPEC strains harbor several pathogenetic attributes that can produce cUTIs in pediatric patients at HIMFG. These strains may be of commensal origin and have a high permissibility of resistance gene acquisition via horizontal gene transfer. Additional whole-genome sequencing studies are needed to verify whether these strains belong to ST131, a highly disseminated clone worldwide. Healthy child carriers of commensal O25b/ST131 strains are widely described, which indicates the relevant dissemination of this clonal lineage worldwide [42,43].

UPEC O25b clinical strains were related to phylogenetic groups B2_1-2_ (69.84%) and D_1-2_ (25.39%) in the present study, which indicates their clinical origin. Phylogenetic group B2 is related to the O25b-B2-ST131 clonal lineage in children and adults. A high percentage of resistance to fluoroquinolones (77.77%), ESBL production (64.28%), and virulence genes were detected in our study, which is similar to previously reported results [30,33,44]. MDR UPEC clinical strains belonging to phylogenetic group D with high virulence potential were identified in Mexican children with cUTIs [15,16]. Phylogenetic group D is related to resistance transmission in *E. coli* strains of the urinary tract and intestinal origin in children and adolescents [45]. Nonpathogenic *E. coli* strains associated with phylogenetic group A are the most prevalent in the human intestinal microbiota, exhibit MDR profiles, are closely correlated with cUTIs, and belong to O25b/ST131 [15,44,46]. In contrast, several non-O25b/ST131 urinary tract *E. coli* strains belonging to phylogenetic groups D, A, and BI are resistant to AMC, CTX, CAZ, cefoxitin, ticarcillin–clavulanic acid, and OFX and carry the cefotaximase CTX-M-15 [47]. These data suggest that the O25b/ST131 clones belonging to phylogenetic groups D and A represent the characteristic acquisition of virulence and resistance, which likely occurred during circulation in the patient’s colonic microbiota, where there is strong selective pressure. Whole-genome sequencing studies must be implemented to understand the relationship of these phylogenetic groups in Mexican pediatric strains.

The genes encoding fimbrial adhesins, such as *papG*II, *csgA*, *fimH*, and *ecpA*, were identified in 80 to 100% of the UPEC O25b/ST131 strains. This result was expected because fimbrial adhesins promote bacterial adherence to the uroepithelium. The FimH and CsgA adhesins participate in the process of bladder epithelium invasion and favor bacterial persistence [16]. Virulence gene diversity is associated with isolation type, phylogenetic group, serotype (O:H), and ST, but the main approach indicates that phylogenetic group B2 strains are highly virulent [44]. UPEC O25b/ST131 clinical strains from the HIMFG harbored >70% carrying *chuA*, >70% carrying *iutD*, and >30% carrying *fyuA*. The gene coding for yersiniabactin is widely distributed in extraintestinal *E. coli* strains of several serotypes and STs [44]. The *hlyA* toxin gene was more prevalent in extraintestinal *E. coli* strains in Spain and France [44], but the *satA* gene was primarily identified in pediatric strains in Mexico. Notably, the genes encoding flagellar motor proteins showed high frequency, i.e., 99.20% for the *motA* and *motB* genes, but the *fliC* gene showed a low distribution (31.74%). Most strains of UPEC O25b/ST131 were motile. However, no correlation was observed between the low frequency of the *fliC* gene and the high percentages of motile UPEC strains, which may be due to the high variability of the *fliC* gene. Flagellar mobility is a necessary factor for urothelium colonization, and it favors bacterial ascent to the renal epithelium together with P fimbria [48]. The expression of flagellar and fimbrial proteins may be inversely coordinated, i.e., highly adherent strains may be nonmotile, and strains with low adherence may be motile [49].

The TosA protein is a nonfimbrial adhesin that mediates adherence to the urinary epithelium, and it is a virulence marker in UPEC strains [49,50]. MDR UPEC strain carriage of the *tosA* gene maintains a high correlation with virulence gene presence [50,51]. Our data showed a low frequency (13.74%) of the *tosA* gene in UPEC O25b-ST131 MDR and XDR strains. The expression of several highly specialized fimbriae (type 1 fimbria, curli, and P fimbria) is required for bacterial adherence to the uroepithelium and renal cells via an on/off system, but the role of TosA in the colonization of the bladder and kidney requires additional study.

High genetic diversity was found among the 126 UPEC O25b/ST131 MDR and XDR strains. These data suggest that the origin of the clinical strains is related to large intestine colonization with commensal strains of *E.*
*coli* O25b/ST131, which enter the bladder due to cross contamination and cause an ascending UTI. ESBL producer O25b/ST131 strains are related to human, animal, and environmental health and food reservoirs [43,52,53,54,55]. Other studies showed UTI-associated strains from the colonic microbiota of the same patient [45]. Regardless of the diversity in O25b/ST131 strains, some PFGE patterns may be more common depending on the geographic area [30]. Our study found that 31.74% of strains showed ≥ 80% similarity, which indicated that 40 of the 126 strains were highly related in the pediatric population in the absence of direct patient contact. The commensal microbiota is an important reservoir of resistance and virulence genes, which may be easily transferred to pathogenic bacteria. Therefore, we hypothesize that commensal *E. coli* strains acquire the necessary resistance and virulence factors to become uropathogens and increase the possibility of acquiring a clinical infection.

UPEC O25b clinical strains were distributed in 14 STs; 22 of the strains were ST131 (25.45%) and related to phylogenetic group B2. The great diversity of STs between extraintestinal strains of *E. coli* is related to geographic region, population type, and temporality. Some STs remain constant in a population over time, but others are highly mobile [56].

ST73, ST93, ST120, ST405, and ST998 were also identified as O25b strains of pediatric origin. Phylogenetic group D with ST131 and A1 with ST131 were identified among the UPEC strains with an MDR phenotype and a high virulence profile. These results indicate that ST131 UPEC strains may spread to other populations with less frequent phylogenetic groups, which is of great importance in Mexico [16]. Most of the O25b strains of phylogenetic group D belonged to ST69 and CC69 in our study, which have spread worldwide and were identified in significant quantities among intestinal commensal *E. coli* strains [57]. The ST69 clone is associated with fecal colonization in children and adolescents, and it frequently exhibits an MDR profile due to a large conjugative plasmid, which confers resistance to tetracycline and trimethoprim–sulfamethoxazole [45]. Among all of the phylogenetic groups identified among the UPEC O25b strains, phylogenetic group D showed a greater diversity of STs (ST10, ST62, ST73, ST405, ST443, ST117, ST421, ST443, ST998, and ST1177). The ST distribution identified in our study is consistent with a meta-analysis study that identified 20 STs as the most prevalent extraintestinal pathogenic *E. coli* (ExPEC) strains worldwide, including ST131, ST69, ST10, ST405, ST38, ST95, ST73, and ST1117 [56], and these STs were among the UPEC O25b strains from Mexican children.

The search continues for new agents with antibacterial activity, such as essential oils extracted from plants with biological and pharmacological properties [58,59,60,61]. The bioactivity of these oils may make them a viable therapy to reduce the current increase in multidrug resistance in UPEC clinical strains, including the O25b/ST131 serogroup, which represents a serious health problem and a great challenge for public health around the world.

## 5. Conclusions

UPEC O25b strains from Mexican children in the HIMFG had high resistance to seven antibiotic categories characterized by ESBL expression and class 1 integron carriage. The UPEC O25b strains also harbored several virulence genes involved in adherence, motility, toxicity, and iron uptake. The resistance and virulence of the UPEC O25b strains were related to phylogenetic groups B2_2_ and D_1_. High genetic diversity was related to 14 STs and 10 CCs of worldwide importance and distribution, which suggests a commensal origin of the O25b strains. The various pathogenic attributes identified in UPEC O25b strains, including resistance, significantly complicate and reduce the choice of antibiotics for the treatment of UTIs. These studies contribute to the implementation of strategies for the control of MDR UPEC strains associated with UTIs.

## Figures and Tables

**Figure 1 microorganisms-09-02299-f001:**
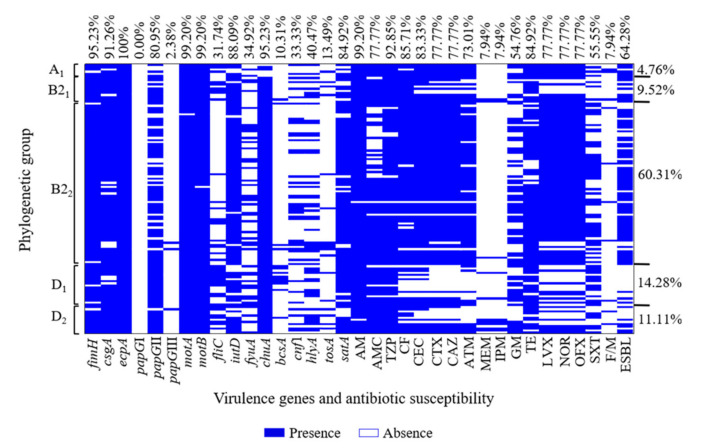
Heat map of virulence genes and antibiotic resistance profiles associated with UPEC O25b strains. Clinical strains of UPEC showing five phylogenetic subgroups: 17 virulence genes (*fimH*, *csgA*, *ecpA*, *papG*I, *papG*II, *papG*III, *fliC*, *motA*, *motB*, *iutD*, *fyuA*, *chuA*, *bcsA*, *cnf1*, *hlyA*, *tosA*, and *satA*), 17 resistance profiles (AM, AMC, TZP, CF, CEC, CTX, CAZ, ATM, MEM, IPM, GM, TE, LVX, NOR, OFX, SXT, and F/M), and ESBL phenotype.

**Figure 2 microorganisms-09-02299-f002:**
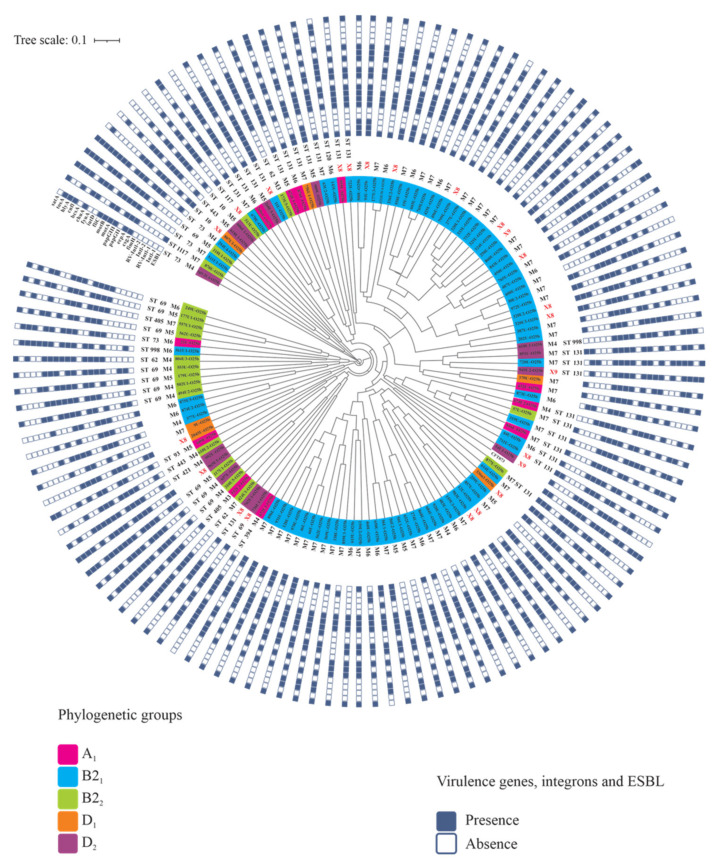
Genetic diversity determined by PFGE of the UPEC O25b clinical strains. The diversity analysis showed a cophenetic correlation coefficient value of r = 0.7917. ST: sequence type, M3: resistance to three antibiotic categories, M4: resistance to four antibiotic categories, M5: resistance to five antibiotic categories, M6: resistance to six antibiotic categories, M7: resistance to seven antibiotic categories, X8: resistance to eight antibiotic categories, and X9: resistance to nine antibiotic categories.

**Figure 3 microorganisms-09-02299-f003:**
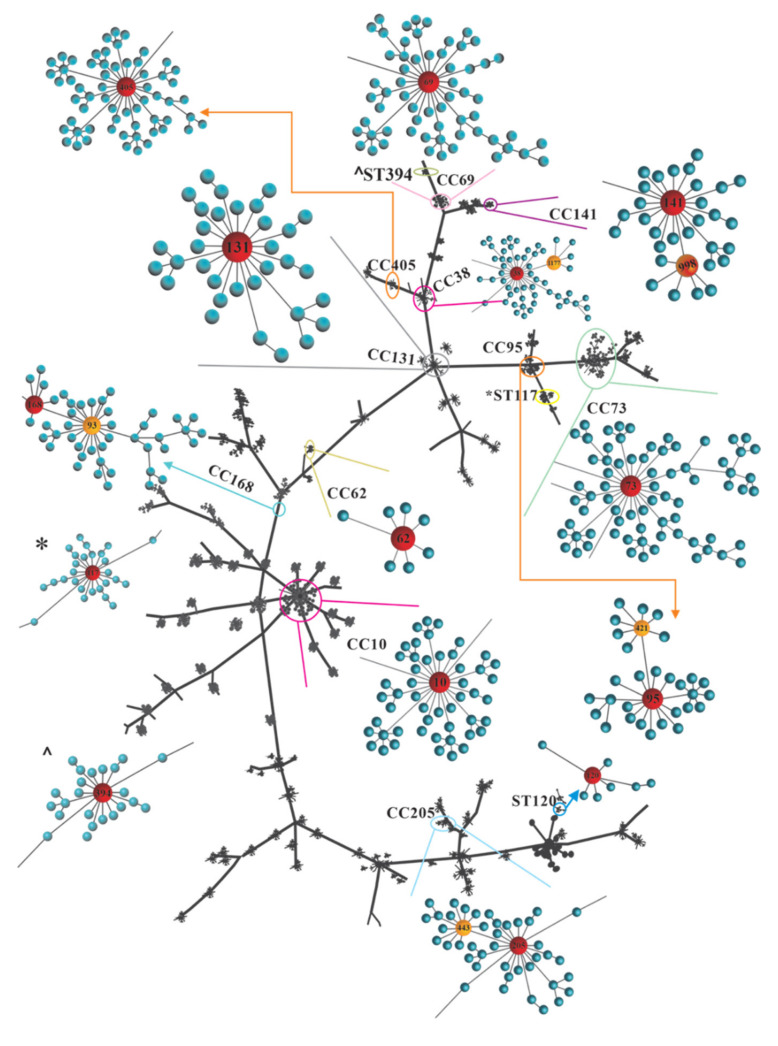
eBURST analysis of UPEC O25b clinical strains. eBURST analysis was performed based on single-locus variants (SLVs). Red shows the founding STs of a clonal complex (CC), and orange shows the STs that differ in only one gene sequence from the founding group.

**Table 1 microorganisms-09-02299-t001:** Resistance to 10 antibiotic categories and ESBL production of 126 UPEC O25b clinical strains.

Antibiotic (µg)	Phylogenetic Group *n* (%)					TR (*n* = 126)
	A_1_ (*n* = 6)	B2_1_ (*n* = 12)	B2_2_ (*n* = 76)	D_1_ (*n* = 18)	D_2_ (*n* = 14)	
Penicillins						
AM-10	6 (100)	12 (100)	76 (100)	18 (100)	13 (92.86)	125 (99.20)
β-Lactam/combination agents						
AMC-20/10	6 (100)	11 (91.7)	52 (68.42)	15 (83.33)	14 (100)	98 (77.77)
TZP-100/10	6 (100)	12 (100)	70 (92.11)	15 (83.33)	14 (100)	117 (92.85)
Cephems 1^st^ and 2^rd^						
CF-30	6 (100)	12 (100)	71 (93.42)	8 (44.44)	11 (78.57)	108 (85.71)
CEC-30	6 (100)	9 (75)	73 (96.05)	8 (44.44)	9 (64.29)	105 (83.33)
Cephems 3^nd^						
CTX-30	6 (100)	8 (66.7)	72 (94.74)	4 (22.22)	8 (57.14)	98 (77.77)
CAZ-30	6 (100)	8 (66.7)	72 (94.74)	4 (22.22)	8 (57.14)	98 (77.77)
Monobactams						
ATM-30	6 (100)	7 (58.33)	68 (89.47)	5 (27.78)	6 (42.86)	92 (73.01)
Carbapenems						
MEM-10	0 (0)	2 (16.7)	1 (1.31)	2(11.11)	5 (35.71)	10 (7.94)
IPM-10	0 (0)	2 (16.7)	1 (1.31)	2 (11.11)	5 (35.71)	10 (7.94)
Aminoglycosides						
GM-10	4 (66.7)	5 (41.7)	42 (55.26)	11 (61.11)	7 (50)	69 (54.76)
Tetracyclines						
TE-30	6 (100)	10 (83.33)	65 (85.52)	15 (83.33)	11 (78.57)	107 (84.92)
Fluoroquinolones						
LVX-5	6 (100)	6 (50)	73 (96.05)	6 (33.33)	7 (50)	98 (77.77)
NOR-10	6 (100)	6 (50)	73 (96.05)	6 (33.33)	7 (50)	98 (77.77)
OFX-5	6 (100)	6 (50)	73 (96.05)	6 (33.33)	7 (50)	98 (77.77)
Folate pathway antagonists						
SXT-1.25/23.75	3 (50)	3 (25)	44 (57.87)	13 (72.22)	7 (50)	70 (55.55)
Nitrofurans						
F/M-300	2 (33.3)	1 (8.33)	4 (5.26)	1 (5.56)	2 (14.29)	10 (7.94)
ESBLs						
	5 (83.33)	8 (66.66) *	59 (77.63) ^ +	3 (16.66) *^	6 (42.85) +	81 (64.28)
Integrons						
Int-1 (*intl1*)	4 (66.66)	2 (16.66) °^ꝏ^	37 (48.68) °	10 (55.55) ^ꝏ^	7 (50.00)	60 (47.61)
Int-2 (*intl2*)	1 (16.66)	1 (8.33)	2 (2.63)	0 (0.00)	0 (0.00)	4 (3.17)

AM-10 (ampicillin), AMC-20/10 (amoxicillin–clavulanate), TZP-100/10 (piperacillin–tazobactam), CF-30 (cefalotin), CEC-30 (cefaclor), CTX-30 (cefotaxime), CAZ-30 (ceftazidime), ATM-30 (aztreonam), MEM-10 (meropenem), IPM-10 (imipenem), GM-10 (gentamicin), TE-30 (tetracycline). LVX-5 (levofloxacin), NOR-10 (norfloxacin), OFX-5 (ofloxacin), SXT-1.25/23.75 (trimethoprim–sulfamethoxazole), F/M-300 (nitrofurantoin), TR (total resistance), Int-1 (integrase class 1), Int-2 (integrase class 2), and 1st, 2nd and 3rd (generation of cephems). Statistically significant: * *p* = 0.0024; ^ *p* < 0.0001; + *p* = 0.0066; ° *p* = 0.0404, and ^ꝏ^
*p* = 0.0380.

## Data Availability

All relevant data are provided in the manuscript.

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
