# Peer review of "Molecular Epidemiology of Multidrug-Resistant Uropathogenic Escherichia coli O25b Strains Associated with Complicated Urinary Tract Infection in Children"

_microorganisms, 2021, doi:10.3390/microorganisms9112299_

Round 1

Reviewer 1 Report

Summary: The authors examined and analyzed the molecular epidemiology in UPEC O25b clinical strains 28 based on their resistant profiles, virulence genes, and genetic diversity. UPEC O25b/ST131 harbors wide genetic diversity, virulence and resistance genes contributing to cUTIs in pediatrics.

Comments: This study has been done using extremely powerful methods and a lot of time and the results and analysis are definitely interesting to  the readers. The reviewer would like to know whether the relationship  could be found between genetic diversity determined by PFGE (or eBURST analysis) and ESBL plasmid types such as CTX-M series, SHV series etc.

Author Response

Revision of manuscript microorganisms-1414657

Comments Suggestions for Authors

Summary: The authors examined and analyzed the molecular epidemiology in UPEC O25b clinical strains 28 based on their resistant profiles, virulence genes, and genetic diversity. UPEC O25b/ST131 harbors wide genetic diversity, virulence and resistance genes contributing to cUTIs in pediatrics.

Comments: This study has been done using extremely powerful methods and a lot of time and the results and analysis are definitely interesting to the readers. The reviewer would like to know whether the relationship could be found between genetic diversity determined by PFGE (or eBURST analysis) and ESBL plasmid types such as CTX-M series, SHV series etc.

Response

In this study, we did not address the molecular determination of resistance associated with plasmid CTX-M type ESBLs, therefore, it is difficult to suggest whether there a relationship with PFGE and MLST. However, in our experience we have not found a relationship between genetic diversity by various methods and the genotypic expression of resistance, associated with the expression of enzymes that hydrolyze the antibiotic.

It is important to comment that the PFGE genetic diversity techniques and CCs by eBURTS are indeed powerful and robust, and together they provide sufficient information on the behavior of microbial populations. These techniques, applied individually, provide little information, considering that each one addresses different aspects. Macrorestriction of the whole genome with an exact cut enzyme in a low number of cuts is the diversity methodology of the PFGE. On the other hand, the analysis of CCs by eBURTS uses the differences in polymorphisms in 7 genes associated with House-keeping type enzymes. In our study, the determination of ESBL and MBL were performed phenotypically.

We want comment that the manuscript was revised, edited, and formatted by the following service https://secure.aje.com/es/researcher/ and with the following verification code 4038-5C82-578C-E59B-F29P. 

Reviewer 2 Report

"Molecular Epidemiology of Multidrug-Resistant Uropathogenic Escherichia coli O25b Strains Associated with Complicated Urinary Tract Infection in Children" is an interesting article with its qualities.

Let's start by stressing the importance of the subject dealt with in it, as are all those related to Public Health, even more when it comes to its child aspect.

It is written in clear and rigorous language, in understandable English. The text often has an encyclopedic air, as well as a report, requiring a very persistent reading on the part of the readers. However, I believe it is difficult, due to the type of subject, to do it differently.

The Introduction is very complete, with abundant reference to relevant concepts and literature. At the end, a brief description of the article's organization is missing.

The "Materials and Methods" section is written with such rigor and detail that it leads us to believe that the research was carried out with great scientific rigor and scrupulousness. The quantitative technique used is suitable for this investigation.

The "Materials and Methods" section is written with such rigor and detail that it leads us to believe that the research was carried out with great scientific rigor and scrupulousness. The quantitative technique used is suitable for this investigation.

The results are very well presented, and exhaustively in Section 3. In "4. Discussion" an exhaustive discussion of the results is made. However, the last periods of this section:

In conclusion, UPEC O25b/ST131 strains from Mexican children in HIMFG have high resistance to multiple antibiotics characterized by ESBLs expression and class 1 integron carriage. Additionally, high genetic diversity suggested a commensal origin of the strains, which were related to 14 STs and 10 CCs of  worldwide importance and distribution.

and integrate a "5. Conclusions" section that would have to be developed further from it. In addition to further developing the conclusions themselves, it could also include a reflection on the policies to be followed, given this health panorama.

In Figure 1, the caption does not identify it as Figure 1. Please correct.

In the text there are a multitude of acronyms. Please make a list of acronyms so that readers don't have to search for their meaning in the middle of the text.

Author Response

Comments and Suggestions for Authors

"Molecular Epidemiology of Multidrug-Resistant Uropathogenic Escherichia coli O25b Strains Associated with Complicated Urinary Tract Infection in Children" is an interesting article with its qualities.

Let's start by stressing the importance of the subject dealt with in it, as are all those related to Public Health, even more when it comes to its child aspect.

Response: Thank you very much for the comment

It is written in clear and rigorous language, in understandable English. The text often has an encyclopedic air, as well as a report, requiring a very persistent reading on the part of the readers. However, I believe it is difficult, due to the type of subject, to do it differently.

Response: Thank you very much for the comment

The Introduction is very complete, with abundant reference to relevant concepts and literature. At the end, a brief description of the article's organization is missing.

Response: Suggested changes were carried out. Lines, 91-96: The sentence are read as follows “The aim of this study was to evaluate the resistance profile to 10 antibiotics categories, -lactamases phenotypic expression (ESBL and MBL), presence of integrons (class 1 and 2), identification of 17 virulence genes, genetic diversity by PFGE (macrorestriction patterns) and multilocus sequence typing (MLST; STs and CCs) in a set of clinical strains of serogroup O25b UPEC recovered from children with cUTIs from the HIMFG.

The "Materials and Methods" section is written with such rigor and detail that it leads us to believe that the research was carried out with great scientific rigor and scrupulousness. The quantitative technique used is suitable for this investigation.

Response: Thank you very much for the comment

The results are very well presented, and exhaustively in Section 3.

Response: Thank you very much for the comment

In "4. Discussion" an exhaustive discussion of the results is made. However, the last periods of this section:

In conclusion, UPEC O25b/ST131 strains from Mexican children in HIMFG have high resistance to multiple antibiotics characterized by ESBLs expression and class 1 integron carriage. Additionally, high genetic diversity suggested a commensal origin of the strains, which were related to 14 STs and 10 CCs of worldwide importance and distribution.

and integrate a "5. Conclusions" section that would have to be developed further from it. In addition to further developing the conclusions themselves, it could also include a reflection on the policies to be followed, given this health panorama.

Response: Suggested change was carried out. The conclusion was separated as the section 5 and the sentences were modified and are read as follows: “ The aim of this study was to evaluate the resistance profile to 10 antibiotics categories, -lactamases phenotypic expression (ESBL and MBL), presence of integrons (class 1 and 2), identification of 17 virulence genes, genetic diversity by PFGE (macrorestriction patterns) and multilocus sequence typing (MLST; STs and CCs) in a set of clinical strains of serogroup O25b UPEC recovered from children with cUTIs from the HIMFG”.

In Figure 1, the caption does not identify it as Figure 1. Please correct.

Response: The figure 1 was modified, which was added the percentages of all genes illustrated.

In the text there are a multitude of acronyms. Please make a list of acronyms so that readers don't have to search for their meaning in the middle of the text.

Response: A list of acronyms was included at the end of the manuscript as was suggested for the reviewer.

Finally, we want comment that the manuscript was revised, edited, and formatted by the following service https://secure.aje.com/es/researcher/ and with the following verification code 4038-5C82-578C-E59B-F29P. 

Round 2

Reviewer 1 Report

The paper is qualified to be published.

Author Response

Response to Academic Editor

"Article that can go through the review process but the discussion needs to be deepened:
  - insert the use of molecules of natural origin against Multidrug-Resistant Uropathogenic Escherichia coli  , being a topical topic that should be explored in the future in uropathogenic strains. Use the following references: PMID: 32344551 ; PMID: 32570731 ; PMID: 33031096 ; PMID: 32260297

Response 1:

Page 12, lines 455 to. 459. The suggestions were made as suggested by the academic editor, including new information to the manuscript. The statements included in the manuscript read as follows: “The search for new agents with antibacterial activity, such as essential oils extracted from plants with biological and pharmacological properties [58-61]. The bioactivity of these oils may be a viable therapy to reduce the current increase of the multidrug-resistance in UPEC clinical strains, including the O25b/ST131 serogroup, which represent a serious health problem and a great challenge for public health around the world”.

Page 15, lines 662 to. 675. The following references were included as suggested by the academic editor.

Trong, Le. N.; Viet, Ho. D.; Quoc, Doan, T.; Tuan, Le, A.; Raal, A.; Usai, D.; Madeddu, S.; Marchetti, M.; Usai, M.; Rappelli, P.; Diaz, N.; Zanetti, S.; Nguyen, H.T.; Cappuccinelli, P.; Donadu, M.G. In vitro Antimicrobial Activity of Essential Oil Extracted from Leaves of Leoheo domatiophorus Chaowasku, D.T. Ngo and H.T. Le in Vietnam. Plants (Basel). 2020, 9(4):453. DOI: 10.3390/plants9040453.

Trong, Le. N.; Viet, Ho. D.; Quoc, Doan, T.; Tuan, Le. A.; Raal, A.; Usai, D.; Sanna, G.; Carta, A.; Rappelli, P.; Diaz, N.; Cappuccinelli, P.; Zanetti, S.; Thi, Nguyen, H.; Donadu, M.G. Biological Activities of Essential Oils from Leaves of Paramignya trimera (Oliv.) Guillaum and Limnocitrus littoralis (Miq.) Swingle. Antibiotics (Basel). 2020, 9(4):207. DOI: 10.3390/antibiotics9040207.

Donadu, M.G.; Trong, Le. N.; Viet, Ho. D.; Quoc, Doan, T.; Tuan, Le, A.; Raal, A.; Usai, M.; Marchetti, M.; Sanna, G.; Madeddu, S.; Rappelli, P.; Diaz, N.; Molicotti, P.; Carta, A.; Piras.; S, Usai. D.; Thi, Nguyen, H.; Cappuccinelli, P.; Zanetti, S. Phytochemical Compositions and Biological Activities of Essential Oils from the Leaves, Rhizomes and Whole Plant of Hornstedtia bella Škorničk. Antibiotics (Basel). 2020, 9(6):334. DOI: 10.3390/antibiotics9060334.

Le, N.T.; Donadu, M.G.; Ho, D.V.; Doan, T.Q.; Le, A.T.; Raal, A.; Usai, D.; Sanna, G.; Marchetti, M.; Usai, M.; Diaz, N.; Rapelli, P.; Zanetti, S.; Cappuccinelli, P.; Nguyen, H.T. Biological activities of essential oil extracted from leaves of Atalantia sessiflora Guillauminin Vietnam. J Infect Dev Ctries. 2020, 14:1054-1064; DOI: 10.3855/jidc.12469.
